Improvement of enzymatic saccharification yield in Arabidopsis thaliana by ectopic expression of the rice SUB1A-1 transcription factor

Núñez-López Lizeth 1 2
Aguirre-Cruz Andrés 3
Barrera-Figueroa Blanca Estela 1 bbarrera@unpa.edu.mx blanca_barrera_f@hotmail.com
Peña-Castro Julián Mario 1 julianp@prodigy.net.mx julianpc@unpa.edu.mx
1 Laboratorio de Biotecnología Vegetal, Instituto de Biotecnología, Universidad del Papaloapan , Tuxtepec, Oaxaca , México
2 División de Estudios de Posgrado, Universidad del Papaloapan , Tuxtepec, Oaxaca , México
3 Taller de Alimentos, Instituto de Biotecnología, Universidad del Papaloapan , Tuxtepec, Oaxaca , México
Mueller-Roeber Bernd
Electronic publication date: 2015 Mar 3
Publication date: 2015
Volume: 3
Electronic Location ID: e817
Received 2014 Dec 21; Accepted 2015 Feb 14
Copyright: © 2015 Núñez-López et al.
Copyright year: 2015
Copyright holder: Núñez-López et al.
License: This is an open access article distributed under the terms of the Creative Commons Attribution License, which permits unrestricted use, distribution, reproduction and adaptation in any medium and for any purpose provided that it is properly attributed. For attribution, the original author(s), title, publication source (PeerJ) and either DOI or URL of the article must be cited.
License URL: https://creativecommons.org/licenses/by/4.0/

Keywords: Bioenergy, Biomass, SUBMERGENCE1, Starch, Bioethanol, Transcription factor, Cell wall

Funding: Secretaría de Educación Pública Consejo Nacional de Ciencia y Tecnología de México Jóvenes Investigadores Ciencia Básica 152643 Blanca Estela Barrera-Figueroa 169619 Programa de Mejoramiento del Profesorado Nuevos Profesores de Tiempo Completo 103.5/11/6720 This work was supported by the Secretaría de Educación Pública, Consejo Nacional de Ciencia y Tecnología de México (http://www.conacyt.mx) Jóvenes Investigadores Ciencia Básica 152643 to Julian Mario Peña-Castro and 169619 to Blanca Estela Barrera-Figueroa and Secretaría de Educación Pública, Programa de Mejoramiento del Profesorado (http://dsa.sep.gob.mx) Nuevos Profesores de Tiempo Completo 103.5/11/6720 to Julian Mario Peña-Castro and Blanca Estela Barrera-Figueroa. Lizeth Núñez-López received a fellowship from the Secretaría de Educación Pública, Programa de Mejoramiento del Profesorado. The funders had no role in study design, data collection and analysis, decision to publish, or preparation of the manuscript.

==============================
Saccharification of polysaccharides releases monosaccharides that can be used by ethanol-producing microorganisms in biofuel production. To improve plant biomass as a raw material for saccharification, factors controlling the accumulation and structure of carbohydrates must be identified. Rice SUB1A-1 is a transcription factor that represses the turnover of starch and postpones energy-consuming growth processes under submergence stress. Arabidopsis was employed to test if heterologous expression of SUB1A-1 or SUB1C-1 (a related gene) can be used to improve saccharification. Cellulolytic and amylolytic enzymatic treatments confirmed that SUB1A-1 transgenics had better saccharification yield than wild-type (Col-0), mainly from accumulated starch. This improved saccharification yield was developmentally controlled; when compared to Col-0, young transgenic vegetative plants yielded 200–300% more glucose, adult vegetative plants yielded 40–90% more glucose and plants in reproductive stage had no difference in yield. We measured photosynthetic parameters, starch granule microstructure, and transcript abundance of genes involved in starch degradation (SEX4, GWD1), juvenile transition (SPL3-5) and meristematic identity (FUL, SOC1) but found no differences to Col-0, indicating that starch accumulation may be controlled by down-regulation of CONSTANS and FLOWERING LOCUS T by SUB1A-1 as previously reported. SUB1A-1 transgenics also offered less resistance to deformation than wild-type concomitant to up-regulation of AtEXP2 expansin and BGL2 glucan-1,3,-beta-glucosidase. We conclude that heterologous SUB1A-1 expression can improve saccharification yield and softness, two traits needed in bioethanol production.

Introduction

Ethanol produced by yeast and bacteria through fermentation of plant-synthesized carbohydrates is one of the oldest biotechnological applications, especially for beverages and food. Production of biological ethanol has emerged as an important means for substitution of traditional hydrocarbon-based fuels (Henry, 2010). Key to successful biofuel production is a net output of energy (Vanholme et al., 2013). The process of bioethanol production is currently under intense research to improve microbial fermentation efficiency, available microbial strains, industrial down- and upstream operations, plant stress tolerance and plant biomass quality (Chundawat et al., 2011; Karnaouri et al., 2013; Vanholme et al., 2013; Ribeiro Reis et al., 2014).

A main goal of plant biomass improvement for biofuel production is an increase in a new trait called saccharification. It is defined as the solubilization of plant carbohydrate reservoirs, mainly starch, cell wall and free sugars (Fig. 1) through physical or enzymatic treatments to yield fermentable carbohydrates (Chuck et al., 2011; Chundawat et al., 2011). In this way, saccharification yield is the amount of fermentable sugars released from starch or cell walls after solubilization per unit of plant biomass (Petersen et al., 2012; Nigorikawa et al., 2012).

Many agricultural relevant plants have high saccharification yields with limited energy input. For example, sugarcane (Saccharum sp.) and sugar beet (Beta vulgaris L.) release a sucrose-rich juice after simple mechanical treatments, which is readily fermentable by microorganisms (Waclawovsky et al., 2010). Potato (Solanum tuberosum L.) tubers and maize (Zea mays ssp. mays L.) seeds require chemical or enzymatic hydrolysis of starch by amylase and amyloglucosidase to release glucose-rich extracts (Bahaji et al., 2013). These two processes are the core of first generation bioethanol production. However, each of these plants has a specific geographical growth range, limited saccharificable tissues (stems, tubers or seeds) and are traditionally employed as food staples, thus raising social and economical concerns (Henry, 2010; Stamm et al., 2012).

Second generation bioethanol production aims to use the abundant cellulose reserves present in agroindustrial waste, grasses and trees to increase plant saccharification yields (Stamm et al., 2012). Drawbacks found in this technology are poor enzymatic saccharification because of complex cell wall architecture, energy-consuming chemical and physical pretreatments for cell wall disruption, multiple genes involved in cell wall synthesis and particular carbon allocation dynamics of each plant developmental stage (Chuck et al., 2011; Chundawat et al., 2011).

Understanding carbon allocation in the plant is the basis of saccharification improvement as a trait of biotechnological interest. During evolution, the use of photosynthetic products in reproduction of wild-plants has developed priority over biomass accumulation; this characteristic must not define final plant architecture in order to breed biofuel crops (Stamm et al., 2012). With the current knowledge of starch metabolism (Streb & Zeeman, 2012; Bahaji et al., 2013), amylopectin architecture (Pfister et al., 2014), tissue-specific carbohydrate usage (Andriotis et al., 2012), cell wall synthesis and deconstruction (Chundawat et al., 2011) and differences between domesticated and wild plants (Bennett, Roberts & Wagstaff, 2012; Slewinski, 2012) it is now possible to test different biotechnological strategies to change carbon allocation and improve raw plant biomass saccharification in the context of first and second generation bioethanol production.

Maize and Arabidopsis thaliana (L.) plants with inducible silencing of genes encoding for starch breakdown enzymes glucan water dikinase (GWD1) and phosphoglucan phosphatase (SEX4) increased starch saccharification yield by 50%–300% when compared to WT (Weise et al., 2012). Increased cellulose saccharification yields of 20%–250% have been achieved in different plant models by expressing peptide inhibitors of pectin synthesis (Lionetti et al., 2010), changing expression patterns of glycosyltransferases involved in xylan synthesis (Petersen et al., 2012), or over-expression of endogenous exoglucanases (Nigorikawa et al., 2012). Mutagenesis has also been applied to isolate Arabidopsis mutants with improved saccharification; while some remained uncharacterized, others were unexpectedly related to disrupted auxin transport (Stamatiou et al., 2013). Starch saccharification yield was increased by over-expressing miRNA156 (Chuck et al., 2011), a factor downstream of the trehalose-6-phosphate (T6P) carbon flux sensing machinery (Wahl et al., 2013).

A plant abiotic stress in which carbohydrate consumption and signaling are crucial for survival is submergence stress. An excess of water around root and aerial organs excludes oxygen from cells, forcing an adjustment from aerobic to anaerobic metabolism (Bailey-Serres & Voesenek, 2008; Lee et al., 2011; Fukao & Xiong, 2013). Plants must finely control the consumption of starch to generate ATP and fuel energy demanding cellular processes because when this reserve is depleted, homeostasis is lost and cell death occurs (Bailey-Serres, Lee & Brinton, 2012).

In rice (Oryza sativa L.), the response in cultivars that have an increased tolerance to flooding stress is mediated by the SUBMERGENCE1 locus (SUB1). SUB1 contains three transcription factors from the Ethylene Response Factor (ERF) Group VII gene family, namely SUB1A-1, SUB1B-1 and SUB1C-1; the main genetic factor for tolerance is SUB1A-1 (Xu et al., 2006). SUB1A-1 mRNA is rapidly induced when plants sense ethylene or low-oxygen conditions and redirects transcription relative to near-isogenic genotypes lacking SUB1A-1 (Jung et al., 2010; Mustroph et al., 2010). Rice varieties that posses SUB1A-1 are more tolerant to flooding stress than plants lacking this gene (Xu et al., 2006). The biochemical mechanism underling this tolerance is that plants express SUB1A-1 during stress and conserve starch and free sugars reserves for longer periods improving survival (Fukao et al., 2006; Fukao, Yeung & Bailey-Serres, 2012). Other roles of SUB1A-1 include the inhibition of cell elongation (Fukao & Bailey-Serres, 2008) and delay of the progression to flowering (Peña-Castro et al., 2011).

When floodwaters recede, SUB1A-1 is down-regulated and normal growth processes resume. Collectively, these molecular and physiological activities leading to effective carbon and energy conservation under submergence to prolong survival are called the Low-Oxygen Quiescence Strategy (LOQS; Bailey-Serres & Voesenek, 2008). When compared to WT, rice plants ectopically expressing SUB1A-1 have a delayed progression to flowering (Fukao & Bailey-Serres, 2008), and constitutive higher free sugars concentration in aerial tissue (mixed stem and leaves) but only show differential starch concentrations under dark-starvation stress (Fukao, Yeung & Bailey-Serres, 2012).

Evolutionary analyses indicate that SUB1A-1 is a descendent of gene duplication and neofunctionalization of SUB1C (Fukao, Harris & Bailey-Serres, 2009; Niroula et al., 2012; Pucciariello & Perata, 2013). However, SUB1C-1 is repressed by SUB1A-1 expression and its presence in rice is not associated with the LOQS. Its up-regulation by submergence, ethylene and GA led to the suggestion that it may be involved in promotion of carbohydrates consumption and cell elongation to enable submerged leaf tissue to grow to the surface of floodwaters (Fukao et al., 2006; Fukao & Bailey-Serres, 2008).

We previously employed Arabidopsis thaliana plants transformed with N-terminal FLAG-tagged 35S: SUB1A-1 (OxSUB1A) to evaluate the recapitulation of LOQS phenotypes observed in SUB1 rice. This confirmed OxSUB1A confers hypersensitivity to ABA, reduces petiole cell elongation associated with hyponastic growth, decreases sensitivity to GA, increases lipid mobilization, and exposed inhibition of flowering as a new integral trait of LOQS (Peña-Castro et al., 2011). In this work, we employed Arabidopsis as a functional prototype to explore if SUB1A-1 over-expression can improve plant biomass saccharification. The rationale was that Arabidopsis plants constitutively expressing SUB1A-1 may also display the LOQS low-starch consumption trait. We also included in the analysis 35S:SUB1C-1 plants (OxSUB1C) to gain further insight on its function.

Materials and Methods

Plant material

Arabidopsis thaliana Col-0 accession was used as the wild-type (WT). Transgenic genotypes were described previously (Peña-Castro et al., 2011). Briefly, SUB1A-1 or SUB1C-1 cDNAs from Oryza sativa cv M202(SUB1) were expressed under Cauliflower Mosaic Virus 35S promoter with a N-terminal immunogenic FLAG-tag in Col-0. Two independent single-copy T4 generation transgenics were used for each transgene: OxSUB1A-L5 and -L12 and OxSUB1C-L6 and -L10.

Plant growth conditions

Arabidopsis seeds were surface sterilized (70% v/v EtOH for 5 min followed by 6% v/v hypochlorite for 2 min and six 1-min rinse steps with ddH2O) and germinated in half-strength Murashige and Skoog agar medium (MS, salts 0.215% w/v, 1% w/v sucrose, 1% w/v agar, pH 5.7) in vertical plates. Seedlings were transferred when 7-day-old to substrate (Sunshine Mix #3 plus 1:4 volume perlite:substrate, autoclaved for 2 h and mixed with 2% w/w slow liberation fertilizer NPK 12:12:17) and watered every 2 days. Germination and growth was under long-day conditions (16 h light/8 h dark, 150 µE m−2 s−1, 60% humidity) in a growth chamber (Conviron CMP6010). ZT0 (Zeitgeber Time) was the start of the light cycle (day). Genotypes were grown side-by-side in a randomized manner to minimize experimental noise.

Reducing sugars, cell wall digestibility and starch content

All rosette leaves were harvested at the time described in each experiment, frozen in liquid nitrogen and stored at −80 °C. For all experiments, leaves were ground to a fine powder in liquid nitrogen with mortar and pestle, weighted and further stored or processed. An experimental strategy was designed to quantify the three main components of saccharification yield, namely free reducing sugars, cell wall digestibility and starch content (Fig. 1).

To measure free reducing sugars, 100–120 mg FW of powdered leaves were incubated with ddH20 for 5 min and centrifuged (13,000 rpm for 5 min) to remove debris. The supernatant (100 µl) was mixed 1:1 with DNS reagent (1% w/v 3,5-dinitrosalicylate, 30% w/v sodium potassium tartrate, 1.6% w/v NaOH) and incubated in a boiling water bath for 5 min, then diluted with 1 ml ddH20 and absorbance was determined at 545 nm in a spectrophotometer (Miller, 1959). A glucose standard curve (0.1 to 5 mg Glucose/ml, R = 0.985) was analyzed and used as reference.

Cellulose digestibility and starch content were enzymatically assayed as previously described (Chuck et al., 2011). To test cellulose saccharification yield, commercial cellulase enzyme complex Accellerase 1500 (Genencor, Cedar Rapids, Iowa, USA) composed of proprietary exoglucanase, endoglucanase, hemicellulase and beta-glucosidase was used. Powdered leaves were weighted in 15 ml capped plastic tubes (100–125 mg FW) and 200 µl of 80% ethanol were added, and the sample vortexed. Next, 3 ml of acetate buffer plus 0.74% w/v CaCl2 (pH 5.0) with 1.7% v/v Accellerase 1500 were added, mixed by inversion and incubated at 50 °C for 24 h with rotation (11 rpm) in an oven. Saccharification was stable from 12–36 h as determined in a preliminary kinetics assay (Fig. S1). Reactions were stopped by incubation at 70 °C for 15 min with rotation in an oven. To measure starch content, samples were treated as described above and further hydrolyzed using the manufacturer’s instructions for the Total Starch K-TSTA kit (Megazyme, Bray, Ireland), which includes a thermostable α-amylase digestion in boiling water for 12 min with vigorous stirring every 4 min, and an amyloglucosidase digestion in a 50 °C water bath for 30 min.

Glucose from cellulose and starch was quantified by glucose oxidase/peroxidase at 510 nm in a spectrophotometer as indicated in the commercial kit Total Starch K-TSTA kit. In parallel, Accellerase buffer (blank), carboximethylcellulose and soluble starch (efficiency probes) were processed. The blank was subtracted from calculations and only experiments with efficiency >93% based on the two probes were employed.

Iodine staining

Starch was visualized in rosettes by iodine staining as previously described (Bahaji et al., 2011; Ovecka et al., 2012) with the following modifications. Whole plants were harvested and immediately infiltrated under vacuum with 3.7% v/v formaldehyde in 0.1 M potassium phosphate buffer (pH 6.6) for ∼10 min. Plants were incubated with hot 80% ethanol for 30 min under constant agitation, stained with iodine solution (KI 2% w/v, I2 1% w/v) for 30 min in the dark and rinsed until the blue precipitate of starch was distinguishable from the yellowish background.

Hardness tests

Fracture properties of leaves were measured using a texture analyzer (Brookfield CT325k; Brookfield Engineering Laboratories, Inc., Middleboro, Massachusetts, USA). The three largest rosette leaves from 23-day-old plants were stacked and placed in a fixture base and perforated in the middle of the left blade (avoiding the central vein) with a puncture test probe for fine films (TA-FSF). Resistance was expressed as the force (Newton) applied to break through the tissue.

Starch granule isolation and scanning electron microscopy

Rosette tissue pulverized in liquid nitrogen (2.5 g) was hydrated in 40 ml of water, sonicated for 10 min (100% power, 20% amplitude, 50% intensity, Hielscher Ultrasonic Processor UP200ST) and centrifuged for 5 min at 4,750 × g. The pellet was washed twice with 50 ml of water, resuspended and filtered through a 100 µm and then a 20 µm membrane. The filtrate was centrifuged again at 4,750 × g and the pellet washed with 20 ml of 100% ethanol. Granules were covered with a gold coat and observed in a scanning electron microscope (Helios NanoLab™ 600; FEI, Hillsboro, Oregon, USA).

Quantitative reverse transcription polymerase chain reaction (qRT-PCR)

Total RNA was extracted from Arabidopsis complete seedlings and qRT-PCR was performed as previously described (Peña-Castro et al., 2011). Primers for TUBULIN2 (TUB2, At1g65480) were previously reported (Wenkel et al., 2006). Primers for EXPANSIN2 (AtEXP2, At5g05290, 5′-TTACACAGCCAAGGCTATGGGCTA-3′ and 5′-GCCAATCATGAGGCACAACATCGT-3′) and GLUCAN-1,3,-BETA-GLUCOSIDASE (BGL2, AT3G57260, 5′-TCCTTCTTCAACCACACAGCTGGAC-3′ and 5′-CCAACGTTGATGTACCGGAATCTGA-3′) were obtained from the AtRTPrimer database (Han & Kim, 2006). Primers for GLUCAN WATER-DIKNASE 1 (GWD1, At1g10760) and STARCH EXCESS 4 phosphoglucan phosphatase (SEX4, At3g52180) were previously reported (Weise et al., 2012). Primers for SQUAMOSA PROMOTER BINDING PROTEIN-LIKE 3 to 5 (SPL3-5, At2g33810 At1g53160 At3g15270), SUPPRESSOR OF OVEREXPRESSION OF CONSTANS 1 (SOC1, At2g45660) and FRUITFUL (FUL, At5g60910) were previously reported (Wahl et al., 2013).

Results and Discussion

Improved production of fermentable sugars in OxSUB1A plants

We analyzed two independent Arabidopsis over-expressing transgenic lines for each SUB1 gene, OxSUB1A-L5 and OxSUB1C-L6 are weak over-expressing lines while OxSUB1A-L12 and OxSUB1C-L10 are strong over-expressing lines (Peña-Castro et al., 2011). As an experimental starting point for analysis of rosettes, we selected Col-0 (Wild-Type, WT) bolting time (22-day-old) since OxSUB1A lines have a late flowering-genotype (Peña-Castro et al., 2011). We hypothesized that flowering inhibition allows OxSUB1A plants to accumulate more carbohydrates at ZT16 (end of day).

To determine starch content and cell wall digestibility in rosette tissue of OxSUB1A and OxSUB1C transgenics, we followed a protocol reported to evaluate saccharification efficiency in switchgrass (Panicum virgatum L.), where plant tissue is digested in two sequential steps (Chuck et al., 2011). In the first reaction, the plant material is used as substrate in a cellulolytic enzymatic cocktail (Accellerase 1500) to breakdown cellulose and hemicellulose into glucose and measure cell wall digestibility. In the second reaction, the tissue is digested with α-amylase and amyloglucosidase to quantify starch. For both digestions, saccharification is expressed as released glucose (Fig. 1).

Figure 1 Experimental strategy.

Experimental strategy followed to quantify the contribution of three different plant carbohydrate pools to saccharification yield.

The carbon pool that yielded most glucose in WT Arabidopsis was starch (8.5 mg of glucose g−1 FW), and then free reducing sugars (0.4 mg of glucose equivalents g−1 FW) followed by cell walls (0.1 mg of glucose g−1 FW).

When only free reducing sugars were determined (no enzymatic treatment), OxSUB1A-L5 had 37% more than WT whereas OxSUB1A-L12 did not show a significant difference (Table 1). If only cellulolytic treatment was applied, an improvement in cell wall saccharification was detected: OxSUB1A-L5 and OxSUB1A-L12 generated 16% and 23% higher yields than WT, respectively (Table 1 and Fig. S1). OxSUB1A-L5 and OxSUB1A-L12 rosette tissue generated 88% and 36% more glucose from starch than WT, respectively (Table 1). These results indicate that ectopic expression of SUB1A-1 allows plants to conserve carbohydrates, mainly starch, under non-stress conditions.

Table 1 Free reducing sugars, cell wall digestibility and starch content of 22-day-old rosettes of Arabidopsis Col-0 and transgenics expressing rice SUB1A-1 or SUB1C-1.

	Free reducing sugara
(mg of reducing
sugars g−1 FW)/% of WT	Cell wall digestibilityb
(mg of glucose g−1 FW)/
% of WT	Starch contentc
(mg of glucose g−1 FW)/
% of WT	
Col-0 WTd	0.40 ± 0.01 a	0.100 ± 0.004 a	8.5 ± 2.7 a	
OxSUB1A L5	0.54 ± 0.02/+37 b	0.116 ± 0.008/+16 b	15.9 ± 0.7/+88 b	
OxSUB1A L12	0.44 ± 0.03/+10 a	0.123 ± 0.010/+23 b	11.6 ± 0.8/+36 c	
OxSUB1C L6	0.42 ± 0.02/+6 a	0.101 ± 0.012/+1 a	10.6 ± 1.1/+25 ac	
OxSUB1C L10	0.33 ± 0.01/−17 c	0.062 ± 0.005/−38 c	7.05 ± 0.5/−7 d	
Notes.

a As measured by Miller’s reagent (dinitrosalicylic acid).

b After 24 h saccharification with Accellerase enzyme mix.

c After amylase/amyloglucosidase digestion.

d Different letters indicate a significant difference between genotypes (P < 0.05, means comparison, Student’s t test). Values are means ± S.E. of three independent experimental replicates, each with n = 5 plants.

It has been reported similar saccharification improvement in switchgrass that over-express miRNA156, a strong inhibitor of the progression to flowering (Chuck et al., 2011). Weak miRNA156 over-expressing lines of switchgrass had better saccharification yield from starch than strongly expressing lines, probably because their growth was less impaired. In this work we also observed that the weak overexpressing line OxSUB1A-L5 had a better saccharification yield than the strong overexpressing line OxSUB1A-L12. It has been recently shown that Arabidopsis ERFs activate strong feedback loops through the enzyme plant cysteine oxidase 1 and 2 (PCO1, PCO2; Weits et al., 2014) and the inhibitory protein hypoxia response attenuator 1 (HRA1; Giuntoli et al., 2014) that when constitutively expressed, lead to suboptimal growth under normal and submergence stress conditions. Therefore, moderate and weak expression of SUB1A-1 is necessary to balance these mechanisms.

These data is consistent with previous research where rice UBI:SUB1A-1 plants had a higher free sugar concentration when measured in aerial tissue (Fukao, Yeung & Bailey-Serres, 2012). However, these rice transgenics do not show a constitutive starch accumulation but the development is severely delayed. The effects of SUB1A-1 on starch accumulation in rice and Arabidopsis may be due to different carbon allocation strategies among monocots and dicots (monocots use stems as storage organ), wild and cultivated plants, environmental cues and development stages (Bennett, Roberts & Wagstaff, 2012; Streb & Zeeman, 2012; Slewinski, 2012; Wang et al., 2013). For example, sex1-1 (gwd) mutants in Arabidopsis accumulate starch and have severe developmental defects since they cannot efficiently match growth and anabolism (Weise et al., 2012; Paparelli et al., 2013), whereas development of rice gwd mutants is normal and only impacts grain yield even when they accumulate up to 400% more starch than WT (Hirose et al., 2013). Overexpression of miRNA156 promoted starch accumulation in switchgrass but not in Arabidopsis, maize or tobacco (Chuck et al., 2011).

When the weak over-expressing line OxSUB1C-L6 was analyzed, it did not show a significant saccharification yield improvement in starch, cell wall or free reducing sugars (Table 1). The strong over-expressing line OxSUB1C-L10 showed decreased saccharification yield for starch and cell wall (−38% and −17%, respectively; Table 1) and lower free reducing sugars levels (−17%; Table 1). These data support the hypothesis that SUB1A-1 and SUB1C-1 control opposing biochemical mechanisms, despite belonging to the same ERF-VII gene family (Fukao et al., 2006; Fukao, Yeung & Bailey-Serres, 2012).

To visualize starch accumulation, we used iodine staining of 14-day-old plants of all transgenic lines and WT. ZT24 was selected as the testing point to increase contrast and observe if accumulation was distinct at the end of the night. The staining showed that both OxSUB1A lines leaves retained more starch in leaves. By contrast, the OxSUB1C lines retained less starch at ZT24 than WT (Fig. 2). Together these biochemical and histological data indicate that maintenance of significantly higher leaf starch is the main contributor to the improved saccharification yield of OxSUB1A plants. Differences in cell wall saccharification and free-sugar content are also distinct from WT but are less determining factors.

Figure 2 Iodine staining of 14-day-old rosette leaves at the end of night.

(A) Formaldehyde infiltrated plants. (B) 80% hot ethanol destained plants. (B) Stained plants show starch as a dark-blue precipitate. Black bar is 1 cm.

Diurnal and developmental starch accumulation patterns of OxSUB1A plants

Leaf starch accumulation has a diurnal pattern with a peak at the end of day and consumption during the night (Bahaji et al., 2013; Ortiz-Marchena et al., 2014). To quantify if starch content could be maintained during the diurnal oscillations as suggested by iodine staining (Fig. 2), we collected 21-day-old OxSUB1A and WT plants at the start and middle of both day and night. WT plants accumulated starch in an expected pattern for transitory starch (Ortiz-Marchena et al., 2014) with a peak at the end of day (Fig. 3). OxSUB1A lines had the same normal accumulation pattern but conserved more starch than WT at all points presented. As previously observed at the end of the day, the weakly over-expressing OxSUB1A-L5 significantly doubled starch content relative to WT whereas the strongly overexpressing OxSUB1A-L12 had only 13%–30% more (Fig. 3).

Figure 3 Diurnal oscillation of starch content of 21-day-old Arabidopsis plants expressing rice SUB1A-1.

Upper bar indicates day (open) and night (black) time (16 h day/8 h night). Asterisks indicate a significant difference between genotypes (P < 0.05, Student’s t test). Data were normalized to Col-0 maximum value at the end of the day ZT16 (9.4 mg of glucose g−1 FW). Values are means of three independent experimental replicates, each with n = 5 plants. Error bars are ±S.E.

Plant development and starch accumulation are genetically coordinated, especially during the vegetative phase change marked by the floral transition (Chuck et al., 2011; Yang et al., 2013; Ortiz-Marchena et al., 2014). To investigate the developmental stages where SUB1A-1 can influence starch conservation improvement, we collected leaves at ZT16 at two WT pre-flowering points (adult vegetative, 18 and 21-day-old) and two WT post-flowering points (reproductive stage, 27 and 31-day-old). In WT plants, starch increased as plants reached bolting time and decreased and stabilized after flowering time when cauline leaves begin to contribute to photosynthetic carbon gain (Early et al., 2009). Interestingly, starch content was higher in both pre-flowering OxSUB1A-L5 and -L12 lines (278% ± 23 S.E. and 189% ± 9 S.E.). This difference decreased until all plants had the same starch content after flowering (Fig. 4A). Iodine starch staining at ZT24 of 14, 21 and 28-day-old rosette leaves matched the pattern of improved starch content (Figs. 4B–4D).

Figure 4 Developmental progression of starch accumulation of Arabidopsis plants expressing SUB1A-1.

(A) Plants were grown (16 h day/8 h night) and collected at ZT16. Black and white arrows indicate bolting day (as the number of days when the floral bud was first visible) of Col-0 and OxSUB1A-L5 and -L12 (21 ± 0.1 d, 28.3 ± 0.7 and 26.7 ± 1.2 days, respectively). Different letters indicate a significant difference between genotypes on the same day (P < 0.05, Student’s t test). Data were normalized to 18-day-old Col-0 value at ZT16 (6.1 mg of glucose g−1 FW). Values are means of three independent experimental replicates, each with n = 5 plants. Error bars are ±S.E. (B–D) Iodine staining of Col-0 and OxSUB1A rosette leaves at (B) 14 day, (C) 21 day and (D) 28 day after germination. Black bar is 1 cm.

This evaluation of diurnal and developmental kinetics further supports the conclusion that starch accumulation is responsible for the improved saccharification yield of OxSUB1A plants. The data also indicate that SUB1A-1 is responsible for the starch conservation trait of the LOQS and that this phenotype is regulated in a developmental manner. Two factors involved in this developmental process are likely the flowering transcription factor CONSTANS (CO) and the florigen gene FLOWERING LOCUS T (FT); both transcripts are significantly down-regulated in OxSUB1A rice and Arabidopsis plants, leading to a late transition to reproductive stage even under an inductive flowering photoperiod (Peña-Castro et al., 2011).

Until recently, an involvement of CO/FT in starch metabolism was not evident because ft and co mutants accumulate similar levels of starch as WT when grown under continuous light; however, mutants of GIGANTEA (GI), an upstream circadian regulator of CO, are strong starch hyperaccumulators (up to 300% of WT levels) (Eimert et al., 1995). Recently, the role of photoperiod in starch accumulation during the floral transition was studied and demonstrated that CO controls starch granule structure via differential diurnal DNA-binding patterns and developmental and diurnal regulation of GRANULE BOUND STARCH SYNTHASE (GBSS; Ortiz-Marchena et al., 2014). Through these events, CO promotes accumulation of starch granules with a higher amylose:amylopectin ratio that can be readily digested proposed to enable a carbohydrate burst that create an optimum metabolic state for flowering. With these results, we hypothesize that down-regulation of CO/FT by SUB1A-1 allows OxSUB1A transgenics to conserve starch that would be otherwise employed for developing inflorescence structures.

The mechanism of starch content improvement mediated by SUB1A-1

Late flowering has been related to improved starch saccharification by mechanisms other than those directly regulated by CO. For example, in switchgrass engineered to over-express miRNA156, young nodes accumulated more starch than WT mature nodes (Chuck et al., 2011). However, miRNA156 is a repressor of vegetative-reproductive transition through a CO parallel pathway that was recently shown to be connected to T6P (Wahl et al., 2013; Yang et al., 2013), a repressor of starch catabolism through KIN10 signaling (Baena-González et al., 2007; Delatte et al., 2011). Defects in enzymatic starch mobilization also lead to late flowering, starch accumulation and size defects (Streb & Zeeman, 2012; Paparelli et al., 2013).

To obtain insight into mechanisms that are different or parallel to CO regulation of starch accumulation in OxSUB1A transgenics, we measured polyphasic fluorescence rise (OJIP kinetics). This method has been used to detect photosynthetically-improved plants with increased carbohydrate accumulation (Gururani et al., 2012). However, no significant differences were detected between the five genotypes analyzed in this study (Table S1), indicating that neither OxSUB1A nor OxSUB1C transgenics posses photosystem efficiency that differs from WT.

Although starch granule architecture and biosynthesis is not a well-understood process (Fettke et al., 2011; Streb & Zeeman, 2012), altered shape and size have been reported in mutants of starch breakdown enzymes GWD1 (also known as SEX1) and SEX4 (Zeeman et al., 2002) and starch synthase 2 and 3 (SS2, SS3; Zhang et al., 2008). We isolated and examined starch granules architecture by scanning electron microscopy and found that starch from OxSUB1A 21-day-old rosettes had the same size and characteristic ellipsoid-like shape of those of WT leaves of the same age (Fig. S2).

In our previously reported microarray studies at ZT8 of OxSUB1A and OxSUB1C seedlings (Peña-Castro et al., 2011), we did not detect a significant change in accumulation of mRNAs related to starch biosynthesis or catabolism. However, since these genes have a circadian oscillation, mostly peaking after midday (Smith et al., 2004), we searched our datasets for statistical outliers associated with this biological process that were up- or down-regulated and evaluated them in RNA from seedlings samples collected at ZT16 (end of day). We tested transcripts encoding starch degrading enzymes GWD1 and SEX4 between WT and OxSUB1A or OxSUB1C but found not significant difference in expression (Table S2).

Recently, it was demonstrated that in parallel to CO, transcription factors of the SQUAMOSA PROMOTER BINDING PROTEIN gene family (SPL3-5) connect carbohydrate metabolism to the vegetative phase change and also lead to late flowering phenotypes (Wahl et al., 2013). In switchgrass, down-regulation of SPL3-5 by miRNA156 promotes late flowering and improvement of saccharification yield by both amylolytic and cellulolytic treatments, without modulation of CO/FT ortholog transcripts (Chuck et al., 2011), supporting the idea that CO and SPL/miRNA156 are parallel pathways in leaves that impact flowering time (Wahl et al., 2013). To test if delayed vegetative phase change in OxSUB1A is related to SPL3-5, we measured transcripts of SPL3-5 and downstream genes SOC1 and FUL in 7-d-old seedlings at ZT16. These transcripts were also statistical outliers down-regulated in our microarrays. The expression of all these transcripts was similar to that of WT plants suggesting independent activity from SPL/miRNA156.

In addition to CO and SPL/miRNA156, post-translational regulation of starch synthesis enzymes by reactive oxygen species (Lepisto et al., 2013) and T6P signaling through the stress integrating kinase SnRK1 regulate starch levels (Baena-González et al., 2007; Mattos Martins et al., 2013). T6P is of particular interest for further research since microarray studies of submergence stress response in different plants indicate there is a dynamic change in the transcripts of trehalose-6-phosphate synthase and trehalose phosphate phosphatase (Jung et al., 2010; Lee et al., 2011; Narsai & Whelan, 2013; Van Veen et al., 2013; Tamang et al., 2014).

Hardness of OxSUB1A leaves

In earlier transcriptome analysis (Peña-Castro et al., 2011) we found that SUB1A-1 promoted in Arabidopsis the up-regulation of 17 genes associated with modification of the cell wall and/or biotic stress response, including endotransglycosylase (XTR3, XTR6), expansin (AtEXP2) and glucan-1,3,-beta-glucosidase (BGL2; Table S3). This latter gene was the most up-regulated transcript relative to WT in 7-d-old seedlings. In addition to their biological importance, cell wall associated proteins are also of technological interest for the development of bioethanol fuel. They consist of enzymes and proteins that can change the mechanical properties of cell wall polymers (cellulose, hemicellulose, lignin and callose) improving cell wall digestibility and saccharification yields (Arantes & Saddler, 2010).

To evaluate if the expression of cell wall associated genes in our transgenics was correlated with a phenotype with modified mechanical properties, we employed a texture analyzer to measure fracture properties of leaves in 23-day-old rosette leaves. Both OxSUB1A-L5 and OxSUB1A-L12 leaves offered significantly less resistance to fracture than WT (67% ± 18 S.D. and 70% ± 11% S.D., respectively). OxSUB1C lines were not statistically different from WT (Fig. 5A). To confirm expression of BGL2 and AtEXP2 in OxSUB1A and OxSUB1C, RNA from 7-day-old seedlings at ZT8 were tested by qRT-PCR. OxSUB1A-L5 and OxSUB1A-L12 expressed more BGL2 (Fig. 5B) and AtEXP2 transcripts (Fig. 5C). WT and OxSUB1C accumulated similar BGL2 mRNA levels, whereas OxSUB1C-L6 had 3-fold more AtEXP2 than WT; however this was not replicated in OxSUB1C-L10 (Fig. 5C).

Figure 5 Leaf hardness phenotype of OxSUB1A and OxSUB1C transgenics.

(A) Hardness comparison of rosette leaves from Arabidopsis Col-0 and plants expressing rice SUB1A-1 and SUB1C-1 genes was determined by a fracture resistance test on leaves of 23-day-old plants. Different letters indicate significant differences from Col-0 (P < 0.01, Student’s t test). Values are means of n = 7 to 13 plants. Error bars are ±S.D. (B–C) Transcript accumulation in 7-day-old complete seedlings at ZT8 (middle of the day) of Col-0, OxSUB1A and OxSUB1C transgenics. (A) BGL2 transcript, (B) AtEXP2 transcript. Transcript abundance was determined by quantitative RT-PCR and normalized to abundance in Col-0 using TUBULIN2 as reference. Values are means of two independent experiments with three technical replicates each. Different letters indicate significant difference with Col-0 (P < 0.01, Student’s t test). Values are means ± S.E.

BGL2 belongs to a multigene family of hydrolytic enzymes involved in fungal pathogen defense and developmental processes related to callose, a transitory β-1,3-glucan relevant for cell wall maturation (Doxey et al., 2007; Park et al., 2014). The rice response during submergence stress includes the expression of genes associated to pathogen stress, and the presence of SUB1A-1 improves this induction (Jung et al., 2010). In Arabidopsis, SUB1A-1 also promoted the constitutive expression of these genes (Peña-Castro et al., 2011). The biotic stress component of the submergence stress response primes plants to resist the pathogens that may increase their access to plant tissue during submergence (Hsu et al., 2013).

When rice plants are submerged, plants encoding SUB1A-1 induce EXPANSIN transcripts early in the stress and restrict them in later stages to conserve energy (Fukao et al., 2006). Expansins are cell wall morphogenic proteins that allow non-enzymatic loosening of cellulose and make it more accessible for enzymes during cell expansion (Arantes & Saddler, 2010). Our expression analysis indicates that EXPANSIN induction is conserved in OxSUB1A transgenics in non-stress growth conditions (Fig. 5C). AtEXP2 is a GA-responsive EXPANSIN normally active during seed germination (Yan et al., 2014).

Together, these data provide evidence that the expression of cell wall and biotic response associated genes mobilized by SUB1A-1 is correlated to a phenotype with decreased mechanical strength and improved cellulose digestibility.

Growth penalty in OxSUB1A plants

Strong constitutive starch conservation in plants is frequently accompanied with a growth penalty derived from their inability to efficiently use this energy reserve (Chuck et al., 2011; Weise et al., 2012; Paparelli et al., 2013; Pfister et al., 2014). We did not observe in our young vegetative OxSUB1A transgenics such penalty (Fig. S3A); however, we detected reduced size (Figs. 4C and 4D) and dry weight stagnation (Fig. S3B) in both pre-flowering (40% loss) and reproductive plants (65% loss). Weak cell walls also risk the plant to suffer pathogen attacks or suboptimal biomechanics (Nigorikawa et al., 2012; Petersen et al., 2012).

These negative features would compromise the development of industrial applications based on plants with improved saccharification traits. Peña-Castro et al. (2011) reported that SUB1A-1 heterozygous individuals have a larger rosette size than Col-0, normal flowering time and fertility; these characteristics are lost when SUB1A-1 is homozygous indicating that control of gene expression dosage is important to achieve optimal results. A proposed solution to these drawbacks is the use of inducible promoters (Weise et al., 2012) or tissue-specific promoters (Petersen et al., 2012) that allow fine-tuning the expression of saccharification traits.

Conclusion

The economies of both industrialized and developing nations are currently based on fuels obtained from petroleum and other hydrocarbon reserves. Plant biotechnology can help the transition towards renewable sources and make energy extraction a more sustainable activity. In this work we demonstrated that ectopic overexpression of the rice SUB1A-1 gene in Arabidopsis confers phenotypes with desirable traits for bioethanol production (Fig. S4). SUB1A-1 maintained the starch conservation phenotype of LOQS under normal growth conditions, improving the amylolytic saccharification yield. Additionally, up-regulation of cell wall associated transcripts associated with cell wall loosening by SUB1A-1 improved cell walls deconstruction. Additional research focusing on balancing growth penalty and sugar content is needed to further optimize and implement a biotechnological strategy to improve biomass saccharification yield based on the promising SUB1A-1 mediated starch conservation and cell wall digestibility. With this information, we propose heterologous SUB1A-1 expression as a new alternative for plant biomass improvement as raw material for bioethanol production.

Supplemental Information

Figure S1 Glucose yield of OxSUB1A-L5 and Col-0 determined in an Accellerase-only digestion assay over 36 h

Glucose content was measured with glucose oxidase. 22-day-old plants. Values are means of two independent experimental replicates, each with n = 3 plants. Error bars are ±S.E.

Click here for additional data file.

Figure S2 Scanning electron microscope images of starch granules isolated by centrifugation from rosette leaves of 21-day-old Col-0

Black bar is 10 µm.

Click here for additional data file.

Figure S3 Dry weight measurements of Col-0 and representative lines of OxSUB1A and OxSUB1C transgenics

(A) Data of rosette tissue of 16 -day-old plants. (B) Data of above-ground tissue (rosette and cauline leaves, bolts and siliques). Different letters indicate significant difference with Col-0 on the same collection point (P < 0.05, Student’s t test). Values are means of n = 12 plants. Error bars are ±S.D.

Click here for additional data file.

Figure S4 Proposed model for improved saccharification phenotype of OxSUB1A-1 Arabidopsis

(A) Arabidopsis plants growing under normal conditions transit from juvenile to reproductive stage using the carbohydrates generated in photosynthesis as an energy resource for development of inflorescences and seeds. (B) OxSUB1A plants display flowering inhibition and starch conservation two characteristics of the rice LOQS response. The constitutive expression of these phenotypes under normal growth conditions leads to an improvement in amylolytic saccharification. (C) SUB1A-1 induces a set of cell wall associated proteins including expansin (AtEXP2) and glucan-1,3,-glucanase (BGL2) that act to weaken cell wall microfibrils and ease of access of external cellulolytic enzymes used for digestion of cellulose in biofuel production as compared to Col-0. CHO: soluble carbohydrates.

Click here for additional data file.

Table S1 Polyphasic fluorescence rise (OJIP kinetics) of Col-0 and Arabidopsis 23-day-old plants ectopically expressing rice SUB1A-1 and SUB1C-1 genes

Values are means of two independent experimental replicates ±S.D., each with n = 5 plants.

Click here for additional data file.

Table S2 Quantitative PCR of Col-0 and Arabidopsis 7-day-old seedlings ectopically expressing rice SUB1A-1

Values are means of two independent experiments with three technical replicates each ±S.E, n = 25 seedlings.

Click here for additional data file.

Table S3 Expression values of cell wall genes significantly up regulated in 7-day-old complete seedlings of OxSUB1A-L5 when compared to Col-0

Grouped using GO and quantified by ATH1 microarray hybridization (Gene Expression Omnibus accession number GSE27669).

Click here for additional data file.

Supplemental Information 8 Raw data obtained and analysed in this study

Click here for additional data file.

We thank Dr. José Abad, Dr. Jacqueline Capataz, Dr. Sandra del Moral, Eng. Juan Hernández and Dr. Enrique Villalobos (UNPA-Tuxtepec) for sharing equipment, reagents and laboratory space, Ms. Fabiola Hernández and Lic. Héctor López (UNPA-Tuxtepec) for administrative assistance, Dr. Gladis Labrada (IPICYT) for technical assistance with SEM and Prof. Julia Bailey-Serres (UC-Riverside) for OxSUB1 seeds, thoughtful discussions and reviewing the preliminary manuscript.

Additional Information and Declarations

Competing Interests

Author Contributions

The authors declare there are no competing interests.

Lizeth Núñez-López performed the experiments, analyzed the data, wrote the paper, prepared figures and/or tables, reviewed drafts of the paper.

Andrés Aguirre-Cruz analyzed the data, contributed reagents/materials/analysis tools, reviewed drafts of the paper.

Blanca Estela Barrera-Figueroa analyzed the data, contributed reagents/materials/analysis tools, wrote the paper, reviewed drafts of the paper.

Julián Mario Peña-Castro conceived and designed the experiments, performed the experiments, analyzed the data, wrote the paper, prepared figures and/or tables, reviewed drafts of the paper.

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
