# Peer review of "Improvement of enzymatic saccharification yield in Arabidopsis thaliana by ectopic expression of the rice SUB1A-1 transcription factor"

_PeerJ, doi:10.7717/peerj.817_

## Round 0.1 · original submission · Major Revisions

· Academic Editor

Major Revisions

When revising your manuscript, please provide a point-by-point response to the reviewers comments and in particular clarify the following points:
From the Reviewer 1 comments:
• The term “juvenile plants” needs to be clarified.
• Define which rosette leaves were used for the saccharification, starch and cell wall digestibility measurements (all leaves of the rosette, or which leaf numbers)?
• How do you explain the often stronger effects in the moderate overexpressor? Could this be due to developmental changes (see also Reviewer 2)?
• Take care about the statistics issues raised.
• Provide information about the reference gene used.
From the Reviewer 2 comments:
• Modify the introduction to better refer to aspects of carbon partitioning and starch metabolism rather than saccharification. Include relevant references as suggested by the reviewer.
• Adjust the title of your manuscript, as it is slightly misleading (you did not change the enzymatic saccharification process per se).
• Carefully revise your manuscript with respect to the potential effect of developmental stage on carbon / starch metabolism. Remove too hypothetic statements about the connection to CO/FT and flowering.
• You may consider to show the data in Table 1 as a graph, but in my opinion this is not essential to change.

·

Basic reporting

No Comments

Experimental design

No Comments

Validity of the findings

No Comments

Additional comments

The authors previously reported the generation of transgenic Arabidopsis thaliana plants that overexpress (moderately and strongly) two different rice transcription factors (TFs), namely SUB1A-1 or SUB1C-1.

The data presented in the manuscript indicate that the expression of cell wall- and biotic response-associated genes altered by the SUB1A-1 transcription factor is correlated to a phenotype with decreased mechanical strength and improved cellulose digestibility. The SUB1C-1 transcription factor has a less prominent effect or in some measurements appears to have an opposite effect on the parameters determined.

The manuscript is well written, the quality of the English is good.

I have the following points which the authors should consider / clarify in a revised version of the manuscript:
• The following sentence in the Abstract needs modification: „This high saccharification yield was developmentally controlled since juvenile transgenic plants yielded 200-300% more glucose than Col-0.” It is not clear how the situation was in mature plants.
• More generally: how do the author define “juvenile plants”? This is not explained in the manuscript. I am addressing this point here as the juvenile-to-adult phase transition is a genetically controlled process within the vegetative phase of plant development. Therefore: do the authors indeed mean “juvenile” plants (and if so, how did they determine that their plants were still in the juvenile stage?) or do they simply mean young plants? This should be clarified, in particular as they also determined the expression of juvenility-related genes (the SPL genes) in their study.
• Saccharification, starch, cell wall digestibility measurements: the authors used “rosette leaves” for their measurements (line 120). Here they need to be more specific (in particular as there are quite significant differences between leaves). Did they use all leaves of the rosette for their studies, or did they select defined leaf numbers or leaves of defined developmental stages or sizes?
• Chapter 3.5 “Further optimization of SUB1A-1 saccharification yield improvement” does not present new results, it is more a suggestion about what to study in the future. This chapter should go to “Conclusions” (and modified accordingly).
• In various measurements (but not all) the biochemical effects were stronger in the moderate SUB1A-1 overexpressor than in the strong overexpressor (e.g. Figure 3). However, the authors do not present a possible explanation for this observation. What do the authors think is the reason and to what extent does this impact to potential use of SUB1A-1 for the biotechnological engineering of plants with better saccharification characteristics?
• Legend to Figure 4, line 436: what is the “budding day”? Do the authors mean “day of bolting”? If it is bolting day, at which developmental stage of the bolt did they determine it? (often this is done when the bolt has a length of 1 cm).
• Figure 5 B,C: where complete seedlings or only parts (e.g. leaves) used? Please specify.
• In Figures 4 and 5 the authors need to explain what the labels a, b, c indicate. Most likely identical letter indicate no statistically significant difference. However, if this is the case, why are the columns for L5 and L12 labeled with a “b”? The data seem largely different between the two lines. Furthermore: In Figure 5B and 5C (gene expression data) error bars and significance are indicated (by the letter labels), however, the legend states that two experiments were performed twice. Note, that a reliable statistics (including error bars) cannot be made on just two measurements. This needs to be cleared by the authors. Finally, which gene was used as the reference genes, was it TUBULIN2? This is not indicated in the manuscript.

Others:
• Line 49: give full species name for Arabidopsis (as it is mentioned for the first time here).
• Line 182: the Weise et al. 2013 paper is missing in the reference list (or should it be Weise et al. 2012?).

Reviewer 2 ·

Basic reporting

This manuscript presents the functional characterization of SUB1A-1 transcription factor in Arabidopsis. Ectopic overexpression of the rice SUB1A-1 altered the content of sugars, especially starch. The authors proposed the use of this gene as potential biotechnological route for biofuel production. Although the theme is relevant, the way that the data is presented is sometimes confusing. The introduction section is mainly focused on biofuel production. However, the results and discussion section speculates a role of this gene in carbon partitioning. Therefore, the introduction should be more focused on the carbon portioning and starch metabolism rather than saccharification.
Furthermore, the title of the publication also does not reflect the work done. The authors did not improve any enzymatic saccharification methods for the transgenic lines, but they evaluate the content of carbon in different pools that can be used for the generation of biofuels.
In the results, the authors argued that the overexpression line presents decreased mechanical strength and improved cell wall digestibility based on the expression of few cell wall genes. However, no changes in the phenotype of leaf resistance measurement were observed. They did not perform any kind of pre-treatment of the lignocellulosic material to prove their hypothesis.

Although the authors focused on starch content for biofuel production, important references are missing, especially from Alisson Smith and Samuel Zeeman´s groups.

Table 1 could provide a better overview of the results if presented as a Figure/Graph.

Language style
An overall revision of the text would help to improve the quality of the text. Several parts of the text are very repetitive. One example is on the end of the introduction and in the beginning of results and discussion sections. Those paragraphs could be combined on the end of the introduction.
In some parts, the authors could be more specificic. This was the case in
# line 95: “inhibition of growth and other traits..” what traits?
# line 315: “ some mutants..” involved in what? It would be nice if the authors could provide additional information about the mutants.
other examples about the language style:
# line 229 “ rice transgenics” instead of transgenic rice;
# line 212: “ the compartment that yielded most glucose..” compartment gives the idea of “ parts of the cell” and not carbon pool/compound as it is written in the text;
# line 263: “…but conserved more starch than WT at all points tested” , I would use “presented’ or maintained higher starch content than the WT”

Experimental design

Please see comments bellow

Validity of the findings

The performed experiments showed a clear tendency of the increased sugar contents (free sugars, cell wall, and especially starch) in the SUB1A-1 overexpressor. However, it is already known that plants with higher starch content tend to have differences in developmental stages, as it was clear the case of figures 2 and 4, and discussed by the authors. Therefore, it is not clear to me if the differences observed are really due to starch accumulation or under the influence of the developmental stage as it seems the case. One evidence for this is shown in figure 4a, where Col-0 and the overexpressors lines have the same starch amount at the budding day. It is possible that carbon accumulation is a consequence of the slow growth rather than changes in starch metabolism generated by changes in SUB1A-1 levels.
Although the authors proposed that SUB1A would regulate CO/FT resulting in high starch content and delay in flowering, no experimental evidence was shown. The authors touched in several aspects of starch metabolism and regulation of development (e.g., GWD1,T6P, SPL3/miRNA156, etc.), but no clear experimental evidences were presented to prove their hypothesis.

---

## Round 0.2 · accepted · Accept

· Academic Editor

Accept

Line 567: “Different letters indicate significant difference with Col-0” should read “... differences from Col-0”.